# Expression of NUCB2/NESF-1 in Breast Cancer Cells

**DOI:** 10.3390/ijms23169177

**Published:** 2022-08-16

**Authors:** Alicja Kmiecik, Katarzyna Ratajczak-Wielgomas, Jędrzej Grzegrzółka, Hanna Romanowicz, Beata Smolarz, Piotr Dziegiel

**Affiliations:** 1Division of Histology and Embryology, Department of Human Morphology and Embryology, Wroclaw Medical University, 50-368 Wroclaw, Poland; 2Department of Pathology, Polish Mother Memorial Hospital-Research Institute, 93-338 Lodz, Poland; 3Department of Human Biology, Faculty of Physiotherapy, Wroclaw University of Health and Sport Sciences, 51-612 Wroclaw, Poland

**Keywords:** nesfatin-1, nucleobindin-2 (NUCB2), breast cancer, prognostic factors

## Abstract

Recently, the expression of NUCB2/NESF-1 has been linked to tumor development. We report NUCB2/NESF-1 expression and its relation to clinicopathological parameters in breast cancer cells. Immunohistochemical reactions were conducted on 446 cases of invasive ductal carcinoma (IDC) and 36 cases of mastopathy. The expression of NUCB2/NESF-1 was also examined at the mRNA and protein levels in breast cancer cell lines. A statistically significant higher level of NUCB2/NESF-1 in IDC cells was noted compared to that in mastopathy samples. The level of NUCB2 expression in the cytoplasm of IDC cells decreased with the increasing degree of tumor malignancy (G). Higher NUCB2 expression was found in tumors with estrogen receptor (ER)-positive and progesterone receptor (PR)-positive phenotypes compared to that in estrogen-receptor-negative and progesterone-receptor-negative cases. Moreover, a higher expression was shown in ER(+) and PR(+) MCF-7 and T47D cell lines compared to that in triple-negative MDA-MB-468 and normal human breast epithelial cells. The analysis of the five-year survival rate indicated that a positive NUCB2/NESF-1 expression in tumor cells was also associated with longer patient survival. The study results suggest that NUCB2/NESF1 may play an important role in malignant transformation and may be a positive prognostic factor in IDC.

## 1. Introduction

Breast cancer (BC) is the most common cancer and the leading cause of cancer mortality in women worldwide. The risk of developing BC increases with age. Although BC is rare in young women, a significant increase in the rates has been observed among patients under 40 years of age with active careers and family life [1]. As a result, BC has become a socioeconomic problem worldwide and one of the major healthcare challenges. BC is a very heterogeneous disease. Of note, even tumors with similar clinicopathological characteristics present different biology, behavior, and treatment response [2,3]. There is an urgent need to define new prognostic and predictive markers to make treatment options more personalized and effective.

NUCB2/NESF1 was first identified in the KM3 acute lymphoblastic leukemia cell line as the EF-hand family of calcium-binding protein in the 1990s [4]. In 2006, Oh et al. identified nesfatin-1 (NESF-1) in the hypothalamic nuclei, an 82-amino-acid peptide derived from the cleavage of nucleobindin-2 [5]. As NUCB2/NESF1 and nesfatin-1 are colocalized, these two names are used interchangeably. Several reports indicate that NUCB2/NESF1 is expressed in numerous peripheral organs and tissues such as the stomach, pancreas, reproductive organs, and adipose tissue [6,7,8,9]. The protein has characteristic functional domains, such as a signal peptide, a Leu/Ile-rich region, two Ca^2+^ binding EF-hand domains separated by an acidic amino-acid-rich region, and a leucine zipper. Therefore, it may play a role in many cellular processes [10,11]. The metabolic function of NUCB2 is related to insulin release, adipocyte differentiation, regulation of the endocrine system, stress, and immune and cardiovascular systems [12,13,14,15]. Recently, the function of NUCB2/NESF1 has been linked to tumor development and metastasis. However, the exact role of NUCB2/NESF1 in human malignancies remains unknown.

Generally, a high expression of nucleobindin-2/NESF-1 is associated with poor outcomes in breast, colon, bladder, prostate, gastric, renal, and endometrial cancer [16,17,18,19,20]. It was demonstrated that NUCB2/NESF1 intensified proliferation, invasion, and migration processes in colon, breast, endometrial, papillary thyroid, bladder, and renal cancer cells [19,21,22,23,24,25]. Additionally, it was shown that the inhibition of NUCB2/NESF1 expression in colon cancer cells suppressed the epithelial-mesenchymal transition (EMT)-related molecules, including N-cadherin, E-cadherin, and β-cadherin as well as EMT properties. In turn, NUCB2/NESF1 was also an inhibitor of the proliferation of human adrenocortical carcinoma and ovarian epithelial carcinoma cells [26,27].

NUCB2 interacts with ART-1, which is an integral membrane protein associated with the extracellular tumor necrosis factor (TNF1) receptor (TNFR1) [28]. TNFR1 binds to TNF-1 and can modulate its activity such as induction of apoptosis, necrosis, angiogenesis, immune cell activation, differentiation, and cell migration [29,30]. These processes are crucial for tumor development. NUCB2/NESF1 as the protein with many functional domains may interact with different partners and can be involved in tumor progression. Therefore, NUCB2/NESF1 has become an interesting target for studies in the context of tumor transformation and progression. There are only two reports which demonstrate that nucleobindin-2 is a BC-related protein. Bearing the above in mind, we investigated the relationship between the expression of NUCB2 and clinicopathological parameters in this type of tumor.

## 2. Results

### 2.1. Immunohistochemical Analysis of NUCB2 Protein Expression in BC and Mastopathy

The expression of NUB2 was noted in the cytoplasm in IDC and mastopathy (Figure 1). NUCB2 expression in the cytoplasm of cancer cells was found in 406 IDC cases (91%). A statistically significant higher level of NUB2 expression was found in IDC cancer cells compared to that in mastopathy (*p* < 0.0001).

### 2.2. The Associations between NUCB2 Expression and Clinicopathological Parameters

Statistical analyses showed that the expression of NUCB2 was significantly lower in mastopathy (IRS 1; 1.225) than in cases with G1 (IRS 6.883; 3.84; *p* < 0.01), G2 (IRS 6.675; 3.771; *p* < 0.01), and G3 (IRS 4.232; 3.185; *p* < 0.05) (Figure 2). The analysis of NUCB2 expression with IDC malignancy grade (G) showed that a significantly higher level of NUCB2 expression was observed in G1 and G2 compared to G3 cases (*p* = 0.0001 and *p* < 0.0001, respectively) (Figure 3).

Moreover, it was found that NUCB2 expression was significantly lower in triple-negative BCs (TNBC) than in other IDC subtypes (*p* < 0.0001) (Figure 4). 

A statistically significant higher expression of NUCB2 in IDC cells was observed in ER + and PR+ compared to ER− and PR− (*p* < 0.0001, Figure 5a,b). Moreover, there was a significant positive correlation between cytoplasmic expression of NUCB2 in IDC and the expression of ER and PR in the analyzed cases (r = 0.2279, *p* < 0.0001, r = 0.2552, *p* < 0.0001, respectively; Spearman rank correlation Figure 5c,d). NUCB2 expression in breast tumors was weakly positive correlated with the patient’s age (r = 0.1029, *p* = 0.0327) Moreover, the expression of the protein was higher in elderly patients (age ≥ 66, *p* = 0.0136) (data not shown). No association was found between a positive expression of NUCB2 and tumor size, stage, HER2 protein, and lymph node metastasis (data not shown).

### 2.3. The Associations between NUCB2 Expression and IDC Patient Survival 

The prognostic significance of NUCB2 expression in IDC was analyzed in relation to a five-year survival rate and the overall survival (OS). The analysis of a five-year survival rate indicated that high NUCB2 expression (IRS ≥ 6) in tumor cells was associated with longer patient survival (*p* = 0.0186, Mantel-Cox test, Figure 6). The analysis of the OS data in the group of IDC patients showed that the expression of NUCB2 was not associated with longer OS (*p* = 0.0754, Mantel-Cox).

However, in our study, a multivariate analysis revealed that NUCB2 was not an independent prognostic factor for the five-year survival rate. The univariate analysis showed that a larger primary tumor size (T3–T4) and the advanced stage (III–IV) were significantly associated with a poorer five-year survival rate in the study cohort. Additionally, the multivariate analysis showed that tumor size and stage were independent prognostic factors in our patients (Table 1).

### 2.4. Expression of NUCB2 in Breast Cancer Cell Lines 

NUCB2 expression was also examined in vitro in selected BC cell lines (MCF-7, T47D, SK-BR-3, MDA-MB-468) as well as normal human breast epithelial cells (hTERT-HME1). A real-time PCR analysis revealed a higher expression of NUCB2 mRNA in ER+ and PR+ T47D and the MCF-7 cell line compared to triple-negative MDA-MB-468 and normal human breast epithelial cells hTERT-HME (Figure 7a). NUCB2 mRNA expression of the SKBR-3 cell line (PR− ER−) was also detected at a high level. The analysis of NUCB2 protein levels by Western blot analysis in BC cell lines also showed higher expression in (ER+ PR+) MCF-7 and T47D cell lines (Figure 7b). Weaker bands were detected in the (PR− ER−) SK-BR-3, (TN) MD-MB-468, and normal human breast epithelial cells hTERT-HME (Figure 7b). Immunofluorescence analysis with confocal microscopy showed more intense cytoplasmic reactions in T47D and MCF-7 compared to those in other cell lines (Figure 7c).

## 3. Discussion

NUCB2/NESF-1 is a pleiotropic peptide with many physiological functions [32,33,34]. Its metabolic function includes food intake, glucose metabolism, and the regulation of the immune, cardiovascular, and endocrine systems. Accumulating evidence indicates that NUCB2/NESF-1 is a new cancer-related protein. Recent studies have shown that NUCB2 is overexpressed in breast, bladder, prostate, clear renal cell carcinoma, ovarian, thyroid, endometrial, gastric, and colon cancer cells compared to that in normal tissue [4]. 

In the present study, we are the first to evaluate NUCB2 expression in BC in such a large group of patients (n = 446). To compare the expression level of NUCB2 protein in cancerous and noncancerous tissues, we used 36 cases of mastopathy. The presence of NUCB2 detected by IHC was higher in BC compared to that in mastopathy (control). The NUCB2 protein was found in the nuclei and the cytoplasm in thyroid and gastric cancer, while NUCB2 expression in BC was limited to the cytoplasm [16,25]. The above data are consistent with the findings of Suzuki et al. and Zeng et al. The expression of NUCB2 in BC was found in 50–80% of BC cases [23,35]. In our cohort of patients, NUCB2 was observed in 91% of samples. 

Histological grading is related to tumor characteristics based on the microscopic appearance of abnormal tumor cells and the tumor tissue compared to normal controls [36]. If tumor cells and the tumor tissue are similar to normal cells and the tissue, the tumor is “well-differentiated” (G1). These tumors grow and spread at a slower rate than “moderately differentiated” (G2) or “poorly differentiated” (G3) tumors which have abnormal-looking cells and do not have normal tissue structures [37]. 

In our study, we noticed the inverse relationship between NUCB2 and an increasing malignancy grade of BC cells, while the lowest expression was found in poorly differentiated BC cells (G3). Currently, it is known that NUCB2 positively correlates with the Gleason grade in prostate cancer and the Fuhrman grade in renal cancer [18,38]. Interestingly, Markowska et al. demonstrated that in type I endometrial cancer, the expression of nesfatin-1 was significantly higher in G1 than in G2 and G3 in total (*p* < 0.05) [39]. Our study is the first research that shows the association of NUCB2 with the histological grade in BC.

We have confirmed that NUCB2/NESF-1 expression is positively correlated with the expression of estrogen receptor (ER), but not with HER2 expression, as previously reported by Suzuki et al. ER-positive means that the cells express the estrogen receptor on their surface and grow in response to the hormone estrogen. ER-positive tumors are much more likely to respond to hormone therapy compared to ER-negative tumors [40]. In turn, cancer cells that express HER2 can be more aggressive as this protein is involved in the cell growth. These cancers tend to grow and spread faster than HER2-negative BCs. However, they are much more likely to respond to treatment with drugs that target the HER2 protein [41]. The positive progesterone receptor (PR) of BC is sensitive to progesterone which allows them to grow. Treatment with endocrine therapy inhibits the growth of these cancer cells [42]. We are the first to indicate that NUCB2 also correlates positively with the PR status. It is established that triple-negative phenotype BCs (TNBC) which do not express ER, PR, or HER2 are more aggressive and are characterized by a poor response to standard treatment and a significantly worse prognosis [43]. We examined the expression of NUCB2 protein in TNBC and other molecular types of BC. We found that the expression of NUCB2 was significantly lower in TNBC compared to other BC samples. Bearing in mind the above, we may conclude that the expression of NUCB2/NESF-1 is associated with less aggressive cancer phenotypes. 

Zeng et al. showed that NUCB2 expression in BC tissue was significantly correlated with the extent of nodal invasion and a poor clinical stage [35]. The same observations were made when the expression of NUCB2 was analyzed in relation to clinicopathological parameters in the colon and gastric cells [16,20]. We found no significant association of NUCB2 with the clinical stage, nodal invasion, or tumor size in our cohort of BC samples. 

Little is known about the mechanism of action of NUCB2 in cancer cells. Suzuki et al. demonstrated that the inhibition of NUCB2 with siRNA in MCF-7 and SKBR-3 BC cell lines decreased cell proliferation [23]. However, they did not detect a significant association between NUCB2 status and Ki-67 in clinical samples. Our IHC results showed that NUCB2-/NESF-1 expression in BC correlated weakly negatively with the expression of the Ki-67 antigen but was not statistically important (data not shown). However, recent in vitro studies showed that NUCB2/NESF-1 knockdown with siRNA or shRNA in bladder cancer, glioblastoma, endometrial cancer, and thyroid cancer cell lines resulted in the inhibition of cell proliferation [19,22,25]. Interestingly, the inhibition of NUCB2 with siRNA in colon cancer cells did not affect proliferation [44]. Surprisingly, Ramanjaneya et al. showed that treatment of H295R adrenocortical cells with recombinant nesfatin-1 resulted in decreased proliferative capacity of the cells [26]. Similar observations were made by Xu et al., who revealed that recombinant nesfatin-1 decreased cell proliferation in ovarian cancer in vitro [27]. Moreover, treatment of the endometrial cancer cell line (Ishikawa) with recombinant nesfatin-1 promoted cell proliferation [19]. Interestingly, Ranjan et al. showed that nesfatin-1-treated mice were characterized by facilitated maturation of testes. Treatment with nesfatin-1 resulted in changes in the expression of some proteins involved in proliferation, differentiation, and apoptosis, for example, the proliferating cell nuclear antigen (PCNA), Blc-2, caspase-3 and GLUT-8 [45,46]. Of note, all of these proteins are important for tumor development and progression. Cancer-associated dysregulation of cell proliferation is known to be related to mTOR signaling. Takagi et al. revealed that intense proliferation of the endometrial cancer cell line was the result of increased mTOR phosphorylation by NUCB2 [19]. On the other hand, Xu et al. showed that NUCB2/NESF-1 decreased mTOR phosphorylation and acted as a tumor suppressor in ovarian cancer [27]. To conclude, the role of NUCB2/NESF-1 in cancers is variable and tissue-specific.

We found a relationship between NUCB2 expression levels and five-year survival rate. Patients with higher expression levels had a significantly higher five-year survival rate, which shows that NUCB2 is a positive prognostic factor. However, we revealed that NUCB2 expression was not related to the overall survival. The results differ significantly from those reported by Zeng et al. who demonstrated that BC patients with high NUCB2/NESF-1 expression had a significantly poorer OS. Interestingly, the survival analysis with an online analysis tool conducted on 2032 BC cases indicated that high NUCB2 expression was related to a higher five-year survival rate (Figure 6c) [31]. These conflicting results concerning patient survival highlight a need for further investigation. The results obtained in the in vitro model confirm our findings in the clinical specimens. We revealed that both mRNA and protein levels of NUCB2 were higher in (PR+, ER+) MCF-7 and T47D compared to MDA-MB-468 (TN) BC cell lines and the hTERT-HME1 line of normal human breast epithelial cells. This finding confirms the previous conclusion that NUCB2 is associated with a less aggressive BC phenotype.

## 4. Materials and Methods

### 4.1. Patient Cohort

Tissue specimens from 446 patients with primary BC (invasive ductal carcinoma; IDC) were obtained from the Polish Mother’s Memorial Hospital Research Institute, Lodz, Poland between January 2004 and March 2012. The control samples included 36 mastopathy samples obtained from the 4th Military Teaching Hospital in Wroclaw, Poland. To perform immunohistochemical staining, they were fixed in 10% neutral buffered formalin and embedded in paraffin. The specific clinicopathological data of the patients are presented in Table 2 The study was approved by the Bioethics Committee at the Wroclaw Medical University (no. KB—277/2022), and all patients gave their written informed consent. The following human BC cell lines were used in this study: MCF7, T47D, SK-BR-3, MDA-MB-468 (Cell Line Collection of the Ludwik Hirszfeld Institute of Immunology and Experimental Therapy, Wroclaw, Poland), as well as the normal human breast epithelial cell line hTERT-HME1 (ATCC).

### 4.2. Immunohistochemistry (IHC)

TMA blocks were cut into 4 μm sections. Deparaffinization, rehydration, and epitope retrieval were performed in the EnVision FLEX Target Retrieval Solution using a Pre-Treatment Link Platform (Dako, Via Real Carpinteria, CA, USA). The IHC reactions were performed in Autostainer Link48 (Dako). Endogenous peroxidase was inactivated using the EnVisonTM FLEX Peroxidase-Blocking Reagent (Dako, 5 min). The samples were incubated with the primary antibody against NUCB2 (1/1000, Novus, St. Charles, MO, USA) and then incubated with secondary goat anti-rabbit immunoglobulin antibodies (EnVision FLEX/HRP) for 30 min. The color reaction was obtained using 3,3′-diaminobenzidine tetrachlorohydrate as a peroxidase substrate. The slides were counterstained with EnVison FLEX Hematoxylin (Dako). Evaluation of IHC reactions: Two independent investigators evaluated the IHC reactions under a BX-41 light microscope (Olympus, Tokyo, Japan). The expression of NUCB2 was evaluated using the semi-quantitative IRS scale, according to Remmele and Stegner. The scale takes into account the percentage of cells with a positive reaction (A—0 points—no cells with a positive reaction; 1 point—1–10% cells with a positive reaction; 2 points—11–50%; 3 points—51–80%; 4 points—>80% cells) as well as the intensity of the color reaction (B—0 points—no reaction; 1 point—low intensity; 2 points—moderate intensity; 3 points—strong intensity reaction). The final score represents the product of the two values and falls in the range of 0–12 (A × B). A five-point evaluation scale was used to assess the nuclear expression of Ki-67 (0—no expression, 1 point—>1%–≤10%, 2 points—>10%, ≤25%, 3 points—>25% ≤50%, 4 points—>50%). The status of ER and PR receptors was scored from 0 to 3 points, depending on the percentage of positive cells (0 points—no reactions; 1 point—1–10%; 2 points—11–50%; 3 points—51–100% stained cells). The expression of HER2 receptors was evaluated using a scale that takes into account both the intensity of the membrane reaction and the percentage of positive tumor cells (47).

### 4.3. RNA Isolation, cDNA Synthesis and Real-Time PCR

Total RNA was isolated from cells using an RNeasy plus mini Kit (Qiagen, Copenhagen, Denmark) according to the manufacturer’s recommendations. The protocol included on-column DNAse digestion to remove the genomic DNA. The quantity and purity of RNA samples were assessed by measuring the absorbance at 260 and 280 nm with a NanoDrop-1000 spectrophotometer (Thermo Fisher Scientific, Wilmington, DE, USA). Reverse transcription reactions were performed using a High-Capacity cDNA Reverse Transcription Kit (Applied Biosystems, Foster City, CA, USA). The reactions were performed in triplicates and evaluated by real-time PCR using a 7500 Fast Real-Time PCR System (Applied Biosystems, City, Province, RRID:SCR_014596) and primers and probes of a TaqMan system (Applied Biosystems,). The primers and probes used in the reactions included NUCB2 Hs00172851_m1 for nucleobindin-2 and ACTB Hs99999903_m1 for β-actin (Applied Biosystems,). Thermal cycling conditions were as follows: polymerase activation at 50 °C for 2 min, preliminary denaturation at 94 °C for 10 min, denaturation at 94 °C for 15 s, annealing of primers and probes and synthesis at 60 °C for 1 min for 40 cycles. For quantification, the samples were normalized against the expression of β-actin-encoding mRNA using the ΔΔCT method. 

### 4.4. Confocal Microscopy

For microculture, 600 µL of 2 × 10^4^ cells/mL suspension of cells was set up on slides with Millicell EZ 8-well glass slides (Merck,) and placed in an incubator at 37 °C for 24 h. After the incubation, the cells were fixed using 4% formaldehyde. The slices were incubated at 4 °C overnight with primary specific polyclonal rabbit anti-NUCB2 (1:1000 dilution; code no. NBP2-35072; Novus Biologicals) at 4 °C overnight. Next, the preparations were incubated for 1 h with donkey anti-rabbit secondary Alexa Fluor 568 conjugated antibody (1:2000 dilution; clone, code no.; Invitrogen, Carlsbad, CA, USA) and were mounted using the Prolong DAPI Mounting Medium (Invitrogen, United StatesI co). The observations were made at objective 60×/1.40 oil using Fluoview FV3000 confocal microscopy (Olympus, RRID:SCR_017015) coupled with Cell Sense software (Olympus, RRID:SCR_016238).

### 4.5. Western Blot (WB) Analysis

Cells (1 × 10^6^) were detached using a plastic cell scraper and solubilized in 100 µL of lysis buffer (20 mM Tris–HCl, pH 8.0, 150 mM NaCl, containing 1 mM ethylenediaminetetraacetate (EDTA), 0.5% NP40, and 1 mM phenylmethylsulfonyl fluoride (Roth, Karlsruhe, Germany)). The soluble proteins were quantified by the bicinchoninic method (Sigma, St. Louis, MO, USA). Cell lysates were subjected to vertical sodium dodecyl sulfate–polyacrylamide gel electrophoresis (SDS–PAGE) (10% gel). Separated proteins were transferred to the PVDF membrane, and blotted proteins were incubated with the primary mAb against NUCB2 (Novus) at 4 °C overnight. After washing, the blots were incubated with HRP-conjugated goat anti-rabbit immunoglobulins (Dako) for 1 h at room temperature. 

### 4.6. Statistical Analysis

The Kolmogorov–Smirnov test was used to evaluate the normality assumption of the examined groups. To compare the differences of examined markers’ expression in all patients’ pairs of groups and clinicopathological data, the unpaired *t*-test and the Mann–Whitney test were used. To compare differences between more than two groups, the Kruskal–Wallis and Dunn’s multiple comparison tests were used. Additionally, the Spearman correlation test was used to analyze the existing correlations. The Kaplan–Meier method was used to construct survival curves. To evaluate the analysis of survival, the Mantel-Cox test was performed. A Cox proportional hazards model with forward stepwise selection was used to calculate univariate and multivariate hazard ratios for the study variables. All statistical analyses were performed using Prism 5.0 (GraphPad, La Jolla, CA, USA) and STATISTICA 10 (StatSoft Inc. Tulsa, OK, USA). The results were considered as statistically significant when *p* < 0.05.

## 5. Conclusions

In conclusion, this paper shows for the first time that the NUCB2 protein is a positive prognostic factor for five-year survival in BC.

## Figures and Tables

**Figure 1 ijms-23-09177-f001:**
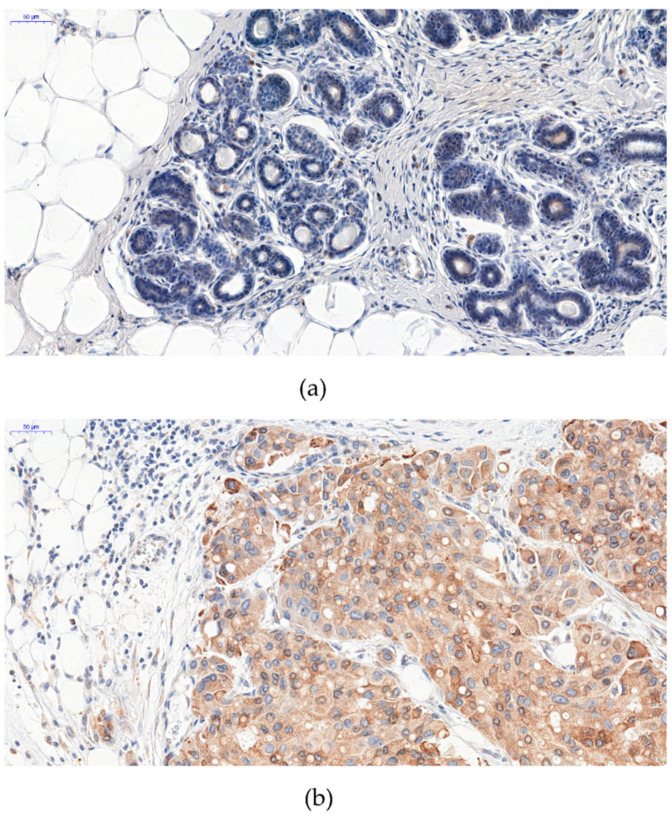
Expression of NUCB2/NESF-1 in mastopathy (**a**) and invasive ductal breast carcinoma (**b**) NUCB2/NESF-1 was located in the cytoplasm of breast cancer cells. Original magnification: 200×.

**Figure 2 ijms-23-09177-f002:**
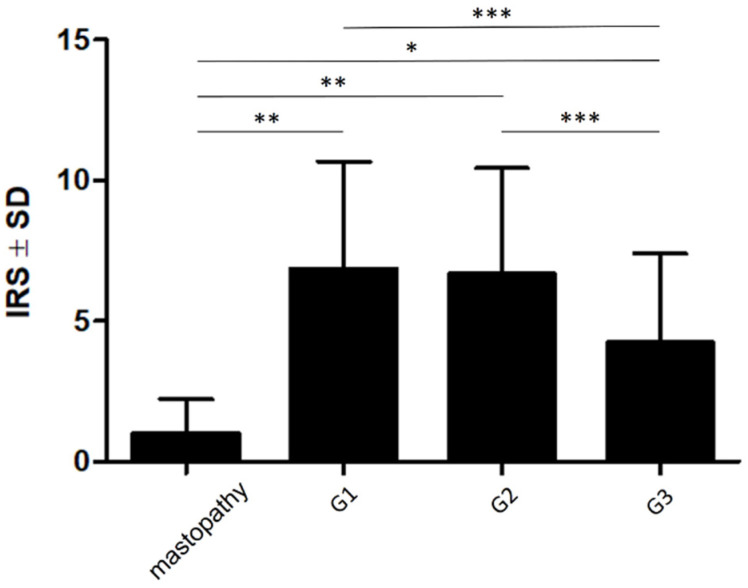
Cytoplasmic expression of nucleobindin-2 in mastopathy in relation to IDC presenting particular malignancy grades. Data presented as the mean ± standard deviation (SD). * *p* < 0.05; ** *p* < 0.01; *** *p* < 0.001.

**Figure 3 ijms-23-09177-f003:**
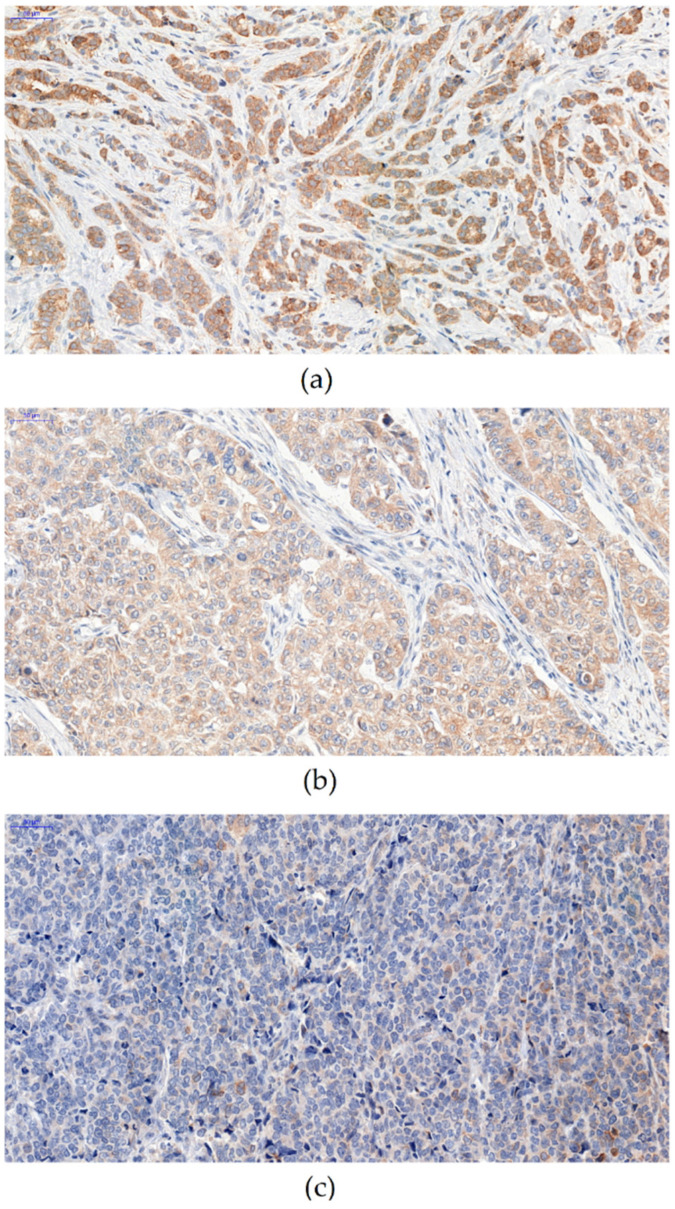
Immunohistochemical expression pattern of NUCB2 in IDC cells. A significantly higher expression level of NUCB2 was observed in G1 (**a**) and G2 (**b**) as compared to G3 (**c**) cases. Magnification 200×.

**Figure 4 ijms-23-09177-f004:**
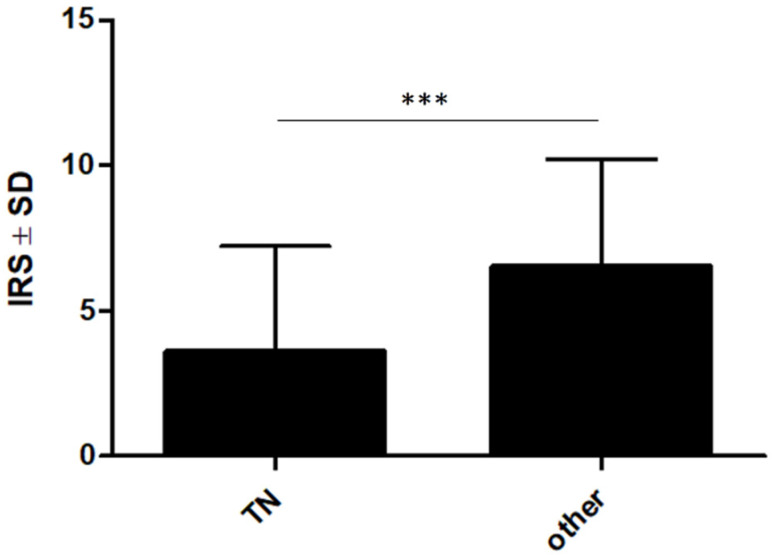
Triple-negative breast cancer (TNBC) cases showed a lower level of NUCB2 expression as compared with that in other molecular subtypes (*** *p* < 0.001).

**Figure 5 ijms-23-09177-f005:**
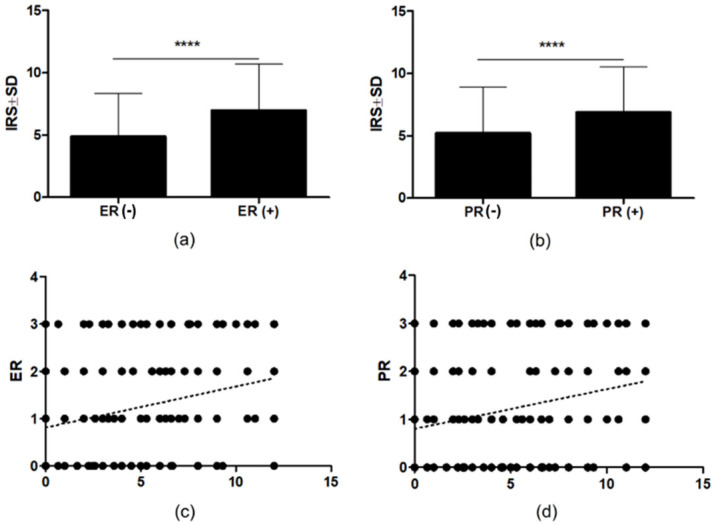
(**a**) ER-positive (ER+) tumors showed a significantly higher level of NUCB2 expression than ER-negative (ER−) cases (**** *p* <0.0001). (**b**) PR-positive (PR+) tumors showed a significantly higher level of NUCB2 expression than PR-negative (PR−) cases (**** *p* <0.0001). (**c**,**d**) A significant positive correlation of cytoplasmic expression of NUCB2 with the expression of ER and PR (r = 0.2779, *p* < 0.0001; r = 0.2552, *p* < 0.0001).

**Figure 6 ijms-23-09177-f006:**
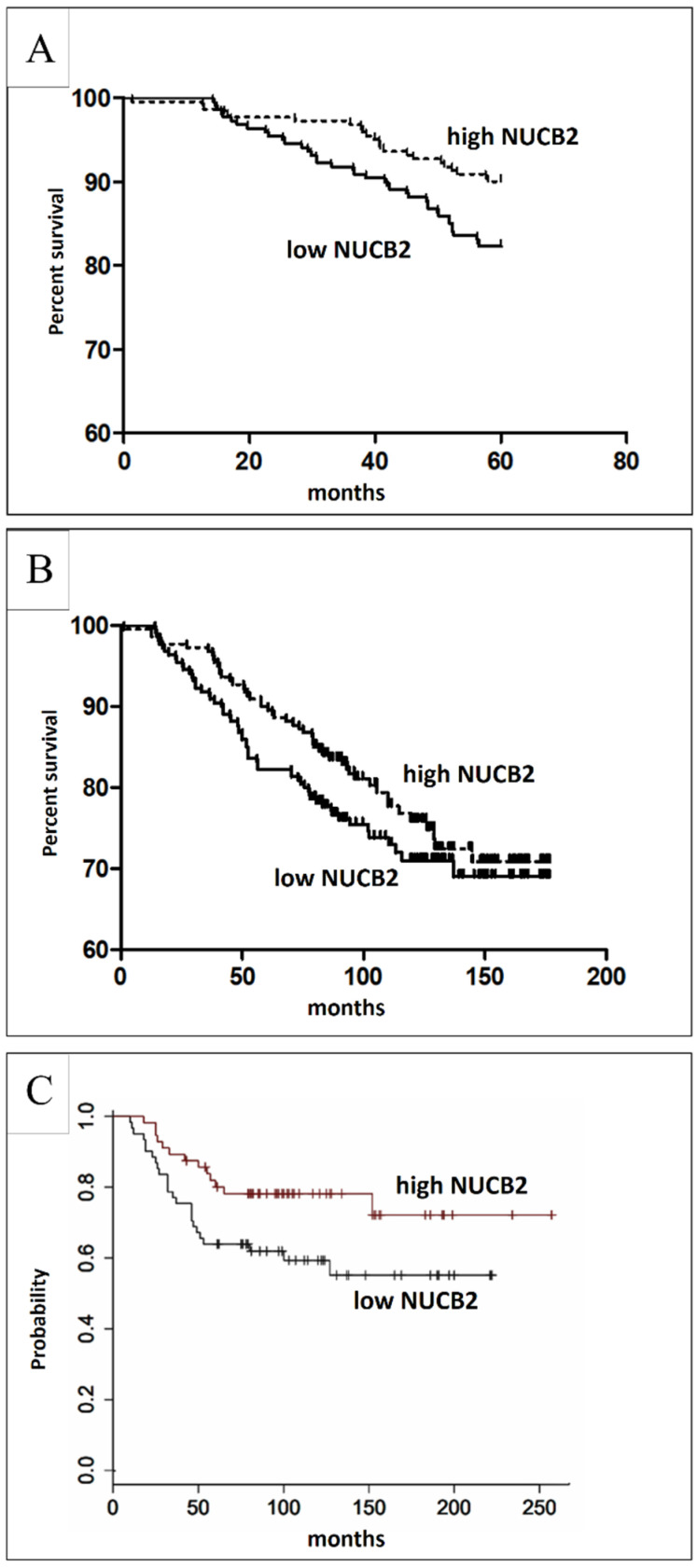
Survival analysis of the NUCB2 protein. (**A**) A higher level of NUCB2 expression strongly correlated with a higher five-year survival rate (*p* = 0.0187). (**B**) NUCB2 expression showed no significant association with overall survival (*p* = 0.2182). (**C**) Survival analysis was performed with an online analysis tool on 2032 cases of BC (*p*= 0.032) [31]. The good prognostic effect of high NUCB2 expression was related to longer overall survival.

**Figure 7 ijms-23-09177-f007:**
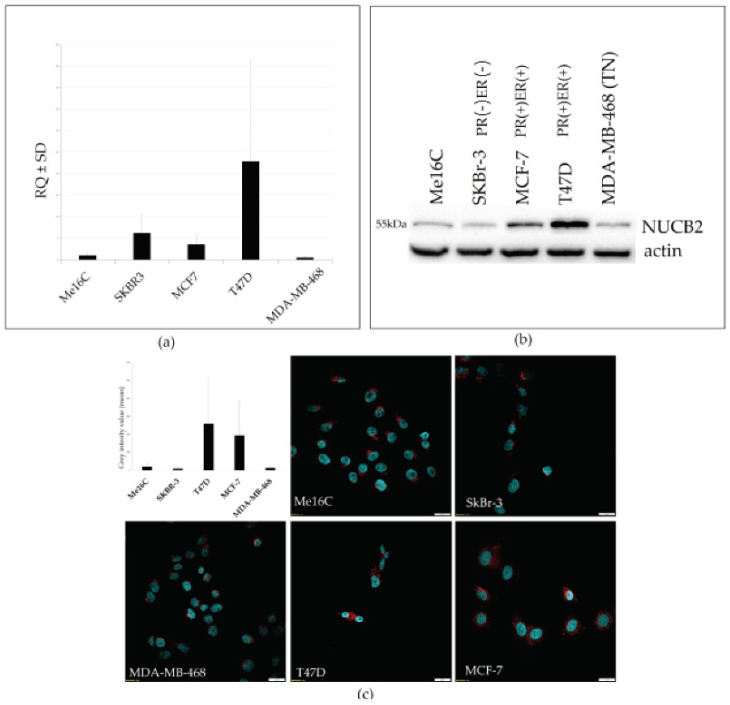
Comparison of NUCB2 expression levels detected by real-time PCR (mRNA) (**a**), Western blot (**b**) and confocal microscopy (**c**) in normal human breast epithelial cell line hTERT-HME1 and different cancer cell lines (MCF7, T47D, SK-BR-3, MDA-MB-468).

**Table 1 ijms-23-09177-t001:** Univariate and multivariate Cox proportional hazards’ analysis of 446 patients with breast cancer.

Clinicopathological Parameters	Breast Cancer	
	Univariate Analysis HR (95% CI) *p*	Multivariate Analysis HR (95% CI) *p*
Grade (G1 vs. G2-G3)	1.02 (0.9–1.0) 0.079196	
pT (1–2 vs. 3–4)	**2.07 (1.5–2.7)** **<0.0001**	**1.68 (1.21–2.33)** **<0.01**
pN (0 vs. 1–3)	1.41 (0.9–2.1) 0.109256	
Stage (1–2 vs. 3–4)	**4.23 (2.3–7.5)** **<0.0001**	**2.15 (1.04–4.45)** **<0.05**
ER (0 vs. 1–3)	0.77 (0.5–1.1) 0.123913	
PR (0 vs. 1–3)	0.72 (0.5–1.0) 0.05778937	
HER-2 (0–2 vs. 3)	1.49 (0.7–2.9) 0.238655	
NUCB2 (median)	0.78 (0.5–1.1) 0.2096233	
NUCB2 (0 vs. 1–12)	0.74 (0.4–1.4) 0.3487976	

Abbreviations: HR—hazard ratio, CI—confidence interval.

**Table 2 ijms-23-09177-t002:** Patient and tumor characteristics of invasive ductal breast cancer (IDC).

Parameters	No.	%
Age (years)		
≤66	239	54
>66	207	46
Tumor size		
T1	259	62
T2	145	35
T3	2	0
T4	9	2
Grade		
G1	66	15
G2	291	68
G3	70	16
Lymph nodes		
N0	255	63
N1, N2, N3	153	38
ER		
Positive	302	68
Negative	142	100
PR		
Positive	294	66
Negative	150	34
TNM		
I	179	44
II	213	52
III	17	4
HER2		
Positive	31	11
Negative	245	89
Triple negative		
Yes	30	7
No	401	93

Missing data: tumor size: 31, grade: 19, lymph nodes: 38, PR:2, ER:2, TNM: 37, HER2:170, triple negative: 15.

## Data Availability

Not applicable.

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
