# Peer review of "Expression of NUCB2/NESF-1 in Breast Cancer Cells"

_ijms, 2022, doi:10.3390/ijms23169177_

Round 1

Reviewer 1 Report

The Kmiecik et al., 2022, Manuscript ID: ijms-1850067 addresses the expression of nesfatin-1 in different breast cancer cell lines. A search on Pubmed.gov for the terms "NUCB2" and "Breast cancer” keywords resulted in very few hits that depicts the novelty of this study.

There are few important queries and few suggestion which makes this manuscript more representable to be publish.

1.       Proper labelling (cell types and of the IHC images and scale bar to show the magnification has to be performed in all the figure in order to attract the relevant readers to read the MS. If possible can the authors provide higher magnification image of Figure 2? The authors are comparing the mastopathy with different grades of tumor but in IHC image, it looks like mastopathy image has been taken in different magnification. The control and Tumor images can be adjusted in 1 figure?

2.       If it is possible can the author supply quantification in IHC images or perform western blot expression data for nesfatin-1?

3.       The authors have not written the molecular size of antibodies used in the figure section.

4.       Do the authors have any medical history of the patients like either they have metabolic disease also and they are taking any medications like metformin?

5.       Five-year survival rate indicated that a positive nesfatin-1 expression in the tumor cells was also associated with longer patient survival. If possible can you check and cite the MS “Immunohistochemically localization and possible functions of nesfatin-1 in the testis of mice during pubertal development and sexual maturation” and “Nesfatin-1 ameliorates type-2 diabetes associated reproductive dysfunction in male mice” consider to incorporate those age groups” by Ranjan et al., 2019; 2021. These papers also deals with the nesfatin-1 and its role in cell survival in discussion section.

6.       Do you have any idea about the changes in the expression of other obesity hormone adiponectin in the same patient slides? It will be very good study or atleast for your future study?

Reviewer 2 Report

The authors analyzed the expression of NUCB2/Nesfatin-1 in relation to clinicopathological parameters in 446 cases of invasive ductal carcinoma and 36 cases of mastopathy using TMA blocks to perform IHC reactions. The results presented in the manuscript are well documented. The statistical analyses are correct. The conclusions are relevant to their results and the search of literature that the role of Nucleobindin-2 peptide in carcinogenesis needs further investigations.

They found a relationship between NUCB2 expression levels and five-year survival rate. Patients with higher expression levels had a significantly higher five-year survival year so that the authors concluded that NUCB2 could be a positive prognostic factor. The search of literature showed conflicting results with opposite effects. They also compared clinicopathological parameters with the expression of NUCB2/NEST-1 and found significant correlation with T size and stage.

NUCB2/NEST-1 is a pleiotropic peptide with many physiological functions. Its metabolic function includes food intake, glucose metabolism, and the regulation of the immune, cardiovascular and endocrine systems. The role of NUCB2/NEST-1 in cancers is variable and tissue-specific. NUCB2/NEST-1 in tumorigenesis seems to be dual – both pro-metastatic and anti-metastatic. The implication of Nubc2/nesfatin-1 seems to be tissue-dependent. Nesfatin-1 was described also as an anorexigenic peptide. This pleiotropic activity of NUCB2/NEST-1 can be the reason of conflicting results obtained in different studies.

Author Response

We would like to thank the Reviewer for his/her time and a constructive review of our study.

Reviewer 3 Report

Dear authors,

Thank you for the opportunity of reviewing the manuscript "Expression of NUCB2/NESF-1 in breast cancer cells". It is a very interesting research since 1994 when NUCB2 was  first described. The group of the studied patients is large enough to ensure statistical significance. They are the first to evaluate NUCB2 expression in BC in such a large group of patients (n=446 ). The researchers also noticed the inverse relationship between NUCB2 and an increasing malignancy grade of BC cells. . Currently, it is known that NUCB2 positively correlates with other cancers such as prostate or renal cancer. They also found a relationship between NUCB2 expression levels and a five-year survival rate.  The studied period is also representative: January 2004 till March 2012. I believe the manuscript is very interesting and worth publication.

Author Response

(The authors gave the same response as above.)

Round 2

Reviewer 1 Report

The authors have tried to justify the queries raised by me.